# Mechanisms Underlying Aboveground and Belowground Litter Decomposition Converge over Time under Nutrient Deposition

**Lei Jiang** [1], **Shenggong Li** [2,3], **Huimin Wang** [3,4], **Xiaoqin Dai** [3,4], **Shengwang Meng** [3,4], **Xiaoli Fu** [3,4], **Jiajia Zheng** [2,3], **Han Yan** [5], **Ning Ma** [2,3], **Yafang Xue** [2,3] and **Liang Kou** [3,4,*]

[1] State Key Laboratory of Subtropical Silviculture, Zhejiang A&F University, Hangzhou 311300, China
[2] National Ecosystem Science Data Center, Key Laboratory of Ecosystem Network Observation and Modeling, Institute of Geographic Sciences and Natural Resources Research, Chinese Academy of Sciences, Beijing 100101, China
[3] College of Resources and Environment, University of Chinese Academy of Sciences, Beijing 100101, China
[4] Qianyanzhou Ecological Research Station, Key Laboratory of Ecosystem Network Observation and Modeling, Institute of Geographic Sciences and Natural Resources Research, Chinese Academy of Sciences, Beijing 100101, China
[5] Plant Ecology, Institute of Biology, Freie Universität Berlin, D-14195 Berlin, Germany
[*] Correspondence: koul@igsnrr.ac.cn; Tel.: +86-010-64889913; Fax: +86-10-64868962

**Abstract:** Decomposition is vital for nutrient cycling and is sensitive to atmospheric nutrient depositions. However, the influences and underlying mechanisms of nutrient deposition on the long-term decomposition of leaves and absorptive roots remain unclear. Here, we explored the responses of leaves and absorptive roots to nutrient deposition (control, +N, +P, and +NP) in *Pinus massoniana* and *Schima superba* forests in subtropical China based on two stages (early-stage (1-year) and late-stage (3-year)) of a decomposition experiment. The chemical factions (acid-unhydrolysable residue (AUR), cellulose, and hemicellulose concentrations) and microbial enzymatic activities (hydrolase and oxidase) were also determined. The +N treatment had persistent negative effects on absorptive root decomposition, except for *P. massoniana* during the late stage. The +P treatment had a positive effect on leaf decomposition in all stages. The +NP treatment had a positive effect on leaf decomposition during the late stage. The increasing decomposition rates of foliar under +P treatment were more correlated with the increasing acid phosphatase activity than chemical factions, indicating a microbial mechanism. The decreasing decomposition rates of roots under +N treatment were weakly correlated with increasing AUR concentrations and strongly correlated with decreasing oxidase activity during the late stage, indicating both chemical and microbial mechanisms. Overall, our findings highlight that, despite contrasting responses to nutrient deposition, the mechanisms underlying aboveground and belowground decomposition tend to converge as decomposition progresses.

**Keywords:** litter decomposition; nitrogen and phosphorus deposition; early- and late-stage; AUR; microbial enzymatic activity; underlying mechanisms

## 1. Introduction

Decomposition is vital for nutrient cycling in terrestrial ecosystems and is sensitive to global environmental change [1,2]. Leaf litter decomposition is a major contributor to soil carbon (C) dynamics, organic matter formation, accumulation, and stability [3,4]. However, a high proportion of plant litter inputs, notably derived from belowground fast-cycling organs, absorptive roots, has frequently been overlooked in the development of research on soil C pools [5,6]. Although some studies have explored absorptive root decomposition [6–8], how long-term ongoing nutrient (e.g., nitrogen (N) and phosphorus (P)) deposition [9,10] drives this process remains unclear, which has impeded its inclusion in global C models. Therefore, a comprehensive understanding of how nutrient deposition

affects decomposition and its underlying mechanisms is crucial for improving the accuracy of ecosystem C and nutrient budgets [11,12].

Evidence has demonstrated the strikingly different effects of increasing atmospheric N deposition on litter decomposition [9,13,14]. This has been ascribed to microbial or chemical mechanisms, depending on litter type [13,15]. For example, a simulated N addition can slow absorptive root decomposition by increasing acid-unhydrolysable residue (AUR) binding to N ions (a chemical mechanism) while having positive, neutral, or negative effects on leaf litter decomposition by regulating microbial enzymatic activities (a microbial mechanism) [9,13]. In contrast to the effects of N deposition on litter decomposition, only a limited number of studies have directly investigated the response of aboveground and belowground litter decomposition to enhanced P deposition, especially in P-limited subtropical forests [9,10,16,17].

Notably, the predominant mechanisms underlying litter decomposition may change over time as the chemical composition of litter and the community structure and enzymatic activity of microorganisms constantly change [18,19]. For instance, labile substances are easily degraded by microorganisms via the secretion of hydrolases [20,21] during the early stages of decomposition. In contrast, macromolecular compounds (e.g., lignin and phenol) can be utilised by microdecomposers that secrete oxidase [22,23] during the later stages of decomposition. Thus, recalcitrant chemical substances and oxidase enzymatic activity may be the main factors explaining the late-stage decomposition under nutrient deposition conditions [23]. It is, however, unclear whether the dominance of chemical and microbial mechanisms change over the course of decomposition under different scenarios of nutrient deposition, particularly with increasing P deposition.

Subtropical forests in China have experienced intense N and P deposition [24,25], which can significantly influence litter decomposition [15]. We previously reported the effects of N and P alone or combined additions on the early-stage (1 year) decomposition of the leaves and absorptive roots of *Pinus massoniana* and *Schima superba* [9]. To explore the long-term effects of nutrient deposition, here, we extended our observation time to late-stage (3 years) decomposition based on the global-scale critical value of leaf litter mass loss (i.e., >40%) [26]. We determined the chemical fraction (AUR, cellulose, and hemicellulose concentrations) and microbial (hydrolase and oxidase) enzymatic activity associated with the two litter types to identify a potential shift in the decomposition mechanisms over time. We hypothesised that: (i) a N addition would inhibit both leaves and absorptive roots decomposition during the late stage, mainly based on a chemical mechanism (i.e., more binding of AUR to inorganic N ions). Simultaneously, we expected that a P addition would stimulate the decomposition of all substrate types via the microbial mechanism (i.e., increasing the P availability can promote the microbial oxidase activity closely related to decomposition during the late stage); (ii) the combined addition of N and P would have less of an effect on litter decomposition, because the positive effect of a P addition would mitigate the inhibitory effect of a N addition; and (iii) the underlying mechanisms controlling decomposition would shift from chemical and hydrolase effects in the early stage to chemical and oxidase effects in the late stage of decomposition.

## 2. Materials and Methods

### 2.1. Site Description

This study took place at the Qianyanzhou Ecological Research Station (26°44′ N, 115°03′ E, 102 m above sea level), Chinese Academy of Sciences, Jiangxi Province, Southeast China. This site has a continental subtropical monsoon climate. According to weather station data, the mean annual temperature and rainfall are 17.9 °C and 1475 mm, respectively. The soil in this study site is categorised as an 'Inceptisol', principally weathering from red sandstone and mudstone [12]. The plantations are dominated by Masson pine (*P. massoniana*), slash pine (*P. elliottii*), *Schima superba*, and Chinese fir (*Cunninghamia lanceolata*). Based on the dominant species and life histories, we selected needle-leaf *P. massoniana* (*Pm*) and broad-leaf *S. superba* (*Ss*) for this study. The current rates of N and P depositions in

this area are approximately 33 N kg ha$^{-1}$ y$^{-1}$ and 0.76 P kg ha$^{-1}$ y$^{-1}$, respectively, which are considered intense deposition rates [25]. More details of the site and the plants were reported by Jiang et al. [9].

## 2.2. Experimental Treatment

In each plantation, a 30 m × 30 m block was designed in January 2016 and then divided into 16 plots (2.5 m × 2.5 m) with at least a 4 m buffer between each plot. The following four treatments were randomly assigned in each block: (1) control (natural conditions, referred to as 'CK'); (2) N-addition (120 kg N ha$^{-1}$ y$^{-1}$, referred to as '+N'); (3) P-addition (40 kg P ha$^{-1}$ y$^{-1}$, referred to as '+P'); and (4) NP-addition (120 kg N + 40 kg P ha$^{-1}$ y$^{-1}$, referred to as '+NP'). Four replicates were randomly selected to receive each of these treatments. Considering the possibility of cross-contamination among each plot within a block, we inserted a rigid plastic baffle (3 m length and 30 cm width) into the soil (15 cm depth) to avoid that. Stand-level fertilisation was initiated in July 2016. The reagent of N (NH$_4$NO$_3$, 35.7 g) and/or P (NaH$_2$PO$_4$, 16.1 g) were fully dissolved in 2 L tap water and applied to the corresponding nutrient-addition plots with a sprayer on rain-free days once every 2 months. The control plots received 2 L of tap water each time. Before beginning the experiment, we sampled soil from the two plantations and measured the background soil properties. The soil nitrate nitrogen, ammonia nitrogen, and available phosphorus were 2.97 ± 0.47 mg kg$^{-1}$, 18.79 ± 0.29 mg kg$^{-1}$, and 4.61 ± 0.55 mg kg$^{-1}$ for *Pm* plantation and 2.14 ± 0.59 mg kg$^{-1}$, 14.16 ± 0.98 mg kg$^{-1}$, and 4.00 ± 0.23 mg kg$^{-1}$ for *Ss* plantation. The total soil C, N, and P concentrations were 18.05 ± 2.16 mg g$^{-1}$, 1.20 ± 0.12 mg g$^{-1}$, and 0.29 ± 0.504 mg g$^{-1}$ for *Pm* plantation and 17.80 ± 3.84 mg g$^{-1}$, 1.18 ± 0.28 mg g$^{-1}$, and 0.26 ± 0.03 mg g$^{-1}$ for *Ss* plantation.

## 2.3. Litter Material Preparation

In May 2016, we used litter traps to collect the freshly dropped intact leaves of *Pm* and *Ss* species. For the belowground root samples, we excavated intact root segments containing at least five branch orders from individual trees and then transported them to the laboratory for no more than 2 h using an incubator. In the laboratory, fine roots were carefully rinsed with deionised water to remove the adhering soil particles or extraneous organic materials. Based on the protocol of fine root classification [27], a pair of forceps was used to dissect the clean, intact, and live fine roots into different hierarchies (i.e., root orders). The 1st and 2nd root orders were used to represent absorptive roots, which were similar to leaves in function. Finally, we categorised the following four substrates, including two species and two fast-cycling organs: *Pm* leaves, *Ss* leaves, *Pm* absorptive roots, and *Ss* absorptive roots. To reduce the influence of the chemical composition of the tissues, we, at a low temperature (40 °C), oven-dried the leaf and root samples to a constant weight before the experiments.

## 2.4. Litterbag Deployment and Retrieval

The litterbag method was used to quantify the decomposition rates in the experimental plots. Litterbags are standard tools in soil ecology [28] and have been used in a large number of experimental field studies [29] despite the recognition that the mesh can prevent natural shredding and mass loss during the in situ decomposition process. Here, 5.0003 ± 0.0002 g dry weight of leaf litter were filled into a 10 cm × 20 cm litterbag (upper surface, 1 mm mesh; lower surface, 0.1 mm mesh), and 2.0003 ± 0.0002 g dry weight of absorptive root litter were filled into a 10 cm × 10 cm litterbag (upper and lower surface, 0.1 mm mesh). The mesh sizes of litterbags can effectively avoid the physical loss of litter substrates [13] and allow access by soil organisms [30]. To simulate the natural conditions of litter decomposition, on 28 July 2016, litterbags containing leaves were put on the soil surface and fixed using nails, whereas litterbags containing roots were completely buried in the soil (10 cm depth). The location of the study area and sampling sites in the *P. massoniana* and *S. superba* plantations are shown in Figure 1.

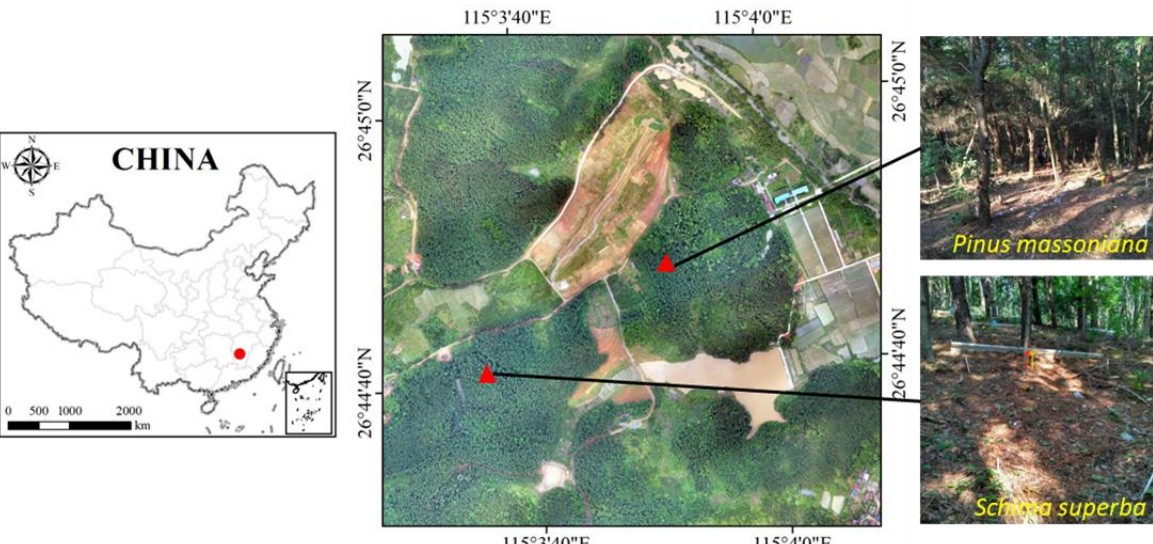

**Figure 1.** Location of the study area and sampling sites in the *Pinus massoniana* and *Schima superba* plantations.

We selected two stages of litterbag retrieval to compare the early- and late-stage effects of the nutrient addition. Thus, litterbags were retrieved on 28 July 2017 (after 1 year) and 28 July 2019 (after 3 years). Intact and undamaged (e.g., animals, pine oil, or other exudates) litterbags were chosen at both sampling occasions. Due to destruction by white ants or turpentine contamination within some blocks, there was an insufficient number of comparable litterbags for determining extracellular enzymatic activities (EEA) at the 3-year interval. Therefore, we collected soil under each decomposing substrate from each plot to determine the EEA. The retrieved samples were immediately transported using an incubator to the laboratory. The attached soil outside the litterbags was carefully cleared before they were opened. The litter was obtained by removing the extraneous materials using a brush and pair of forceps and then oven-dried (40 °C, 48 h) to determine the residual mass.

*2.5. Chemistry Analysis*

For the litter chemical composition, the C and N concentrations were determined with an elemental analyser (Vario MACRO cube; Elementar Analysensysteme GmbH, Langenselbold, Germany). The P concentration was determined by microwave digestion of the powder samples in concentrated $HNO_3$ (ICPMS; Optima 5300 DV, Perkin Elmer, Waltham, MA, USA). The cellulose, hemicellulose, and AUR concentrations were determined using a modified method described by Jiang et al. [9]. Ground subsamples were extracted with benzene-alcohol, and then, the residual part was hydrolysed using the digestion of $H_2SO_4$ (first 72% and second 4%) and filtered. The solid hydrolysate was oven-dried (60 °C, 48 h) to determine the AUR concentrations. The cellulose and hemicellulose concentrations were quantified using high-performance liquid chromatography (UltimateR 3000; Sunnyvale, CA, USA). The ash content of the samples was determined by combusting the samples (40–50 mg) in a muffle furnace (550 °C, 4 h). The values of the chemical traits and litter mass were expressed on an ash-free, dry weight basis.

*2.6. Microbial Extracellular Enzymatic Activity Assay*

The litterbags were sampled at the 1-year stage, and the soil under each retrieval litterbag in each plot was sampled at the 3-year stage to analyse the microbial EEAs. We selected four hydrolytic enzymes and two oxidative enzymes that strongly influenced the litter decomposition in the early and later stages. These hydrolytic enzymes are mainly involved in microbial C, N, and P acquisition, namely *β*-1,4-glucosidase (BG), *β*-1,4-N-

acetylglucosaminidase (NAG), cellobiohydrolase (CBH), and acid phosphatase (AP). For the later (3-year) stage, we selected the same four hydrolytic enzymes plus two oxidative enzymes, peroxidase (PER) and polyphenol oxidase (PPO), which influence the degradation of phenolic compounds, lignin, and aromatic polymers. Before starting the EEAs test, the leaves were cut into approximately 1 cm lengths (*Pm*) or 1 cm$^2$ pieces (*Ss*), and the roots were kept in their original conditions.

For each case, one-half of each residual litter sample, or approximately 20 g of fresh soil, were oven-dried to determine the water content, and 0.5 g of litter or 0.1 g of fresh soil were weighed and shocked with 125 mL of acetate buffer (50 mM, pH 5.0) to prepare the sample, respectively. The sample suspensions, acetate buffer, methylumbelliferyl standard, and substrate solutions were dispensed into 96-well microplates, accompanying eight replicate wells as the treatment. All assays were incubated in the dark (20 °C, 4 h). To stop the reaction, NaOH solution (1 M, 10 mL) was added to each well. Fluorescence at 365 nm excitation and 450 nm emission filters was subsequently measured to achieve the original data (unit, nmol h$^{-1}$ g$^{-1}$).

Oxidase activity was assayed spectrophotometrically. Briefly, the soil suspension (600 μL) and substrate solution (150 μL) were dispensed into deep-well plates. For PER, a H$_2$O$_2$ solution (30 μL, 0.3%) was added to each well. The oxidase plates were incubated in the dark (20 °C, 5 h). The deep-well plates were centrifuged for 3 min, and then, the supernatant was absorbed (250 μL) to measure the enzyme values (unit, nmol h$^{-1}$ g$^{-1}$) at 450 nm using a microplate fluorometer.

### 2.7. Statistical Analysis

The percentage mass loss after 1 and 3 years of decomposition was calculated as follows: mass loss (%) = ($mass_i - mass_t$)/$mass_i \times 100$, where $mass_i$ and $mass_t$ are the dry weight (g) of leaves and absorptive roots at the initial and retrieval times, respectively. The distribution of the values was tested for normality (Shapiro-Wilk test, $\alpha = 0.05$) and log-transformed when necessary for all the measured values. We used analysis of variance (ANOVA) to identify the effects of species and organs on the initial chemical concentrations. Repeated-measures (retrieval time as a factor) ANOVA were conducted to test the effects of different treatments (CK, +N, +P, and +NP) on the mass loss. Two-way ANOVAs were used to test the effects of different treatments on the cellulose, hemicellulose, AUR concentrations, and microbial EEAs among the four substrates. Significant differences were determined using Tukey's honestly significant difference test. Linear regression analyses were used to determine the relationships of 'net' values among the mass loss (after 3 years); residual AUR (after 3 years); and EEA (after 3 years: BG, NAG, CBH, AP, PER, and PPO) among the four substrates. The results were considered statistically significant at $p < 0.05$. Analyses were performed using SAS software (version 9.4; SAS Institute Inc., Cary, NC, USA).

## 3. Results

### 3.1. Initial Chemical Parameters

The absorptive roots litter was relatively low quality, with higher N and AUR concentrations and AUR:P ratios, whereas the leaf litter was relatively high quality, with higher C, cellulose, and hemicellulose concentrations and C:N and N:P ratios (Table 1). Regardless of the substrate type, *Pm* had a lower N concentration and N:P ratio but higher C, AUR, cellulose, and hemicellulose concentrations and lower C:N, AUR:N, and AUR:P ratios than *Ss* (Table 1). The initial chemistry of the four decomposition substrates varied significantly between the species and plant tissue types (Tables 1 and 2). Significant interactions between species and plant tissues were observed for all measured parameters, with the exception of the C:P ratio (Table 2).

**Table 1.** Initial chemistry of the four substrates used in this study.

| Species | Type | C (mg g⁻¹) | N (mg g⁻¹) | P (mg g⁻¹) | AUR (mg g⁻¹) | Cellulose (mg g⁻¹) | Hemicellulose (mg g⁻¹) | C:N | C:P | N:P | AUR:N | AUR:P |
|---|---|---|---|---|---|---|---|---|---|---|---|---|
| *Pinus massoniana* | Leaves | 506.90 [a] (2.83) | 8.07 [c] (0.51) | 0.59 [a] (0.03) | 460.43 [b] (3.62) | 212.40 [a] (1.65) | 201.34 [a] (2.91) | 63.50 [a] (3.49) | 874.12 [a] (54.13) | 13.83 [c] (0.82) | 57.66 [a] (3.09) | 793.72 [b] (47.86) |
| | Absorptive roots | 426.68 [c] (7.19) | 11.88 [b] (0.43) | 0.55 [a] (0.01) | 589.13 [a] (9.91) | 90.71 [c] (2.70) | 112.44 [c] (5.70) | 36.06 [b] (1.52) | 782.46 [a] (24.91) | 21.77 [b] (0.88) | 49.77 [a] (1.93) | 1080.25 [a] (33.06) |
| *Schima superba* | Leaves | 436.43 [c] (3.39) | 15.13 [a] (0.45) | 0.56 [a] (0.02) | 381.74 [c] (10.54) | 183.85 [b] (5.45) | 174.29 [b] (4.78) | 28.92 [b] (0.86) | 788.63 [a] (31.99) | 27.26 [a] (0.67) | 25.31 [b] (1.12) | 690.80 [b] (39.71) |
| | Absorptive roots | 463.44 [b] (2.63) | 14.59 [a] (0.28) | 0.63 [a] (0.01) | 429.14 [b] (4.46) | 201.12 [a] (2.81) | 179.33 [ab] (7.22) | 31.79 [b] (0.55) | 734.07 [a] (10.45) | 23.10 [b] (0.36) | 29.43 [b] (0.39) | 679.68 [b] (10.16) |

Values are presented as the mean with standard error in parentheses. Ratios are mass-based. Significant differences between means were determined using Tukey's honestly significant difference test. Different letters in a column indicate significant differences among treatments ($n = 4$, $p < 0.05$). AUR, acid-unhydrolysable residue.

**Table 2.** Results (*p*-values) of the two-way ANOVA showing the effects of species, tissue, and their interactions on the initial chemical properties of four decomposition substrates.

| Source of Variation | C (mg g⁻¹) | N (mg g⁻¹) | P (mg g⁻¹) | AUR (mg g⁻¹) | Cellulose (mg g⁻¹) | Hemicellulose (mg g⁻¹) | C:N | C:P | N:P | AUR:N | AUR:P |
|---|---|---|---|---|---|---|---|---|---|---|---|
| Species | 0.003 | <0.001 | 0.199 | <0.001 | <0.001 | 0.003 | <0.001 | 0.074 | <0.001 | <0.001 | <0.001 |
| Tissue | <0.001 | 0.002 | 0.396 | <0.001 | <0.001 | <0.001 | <0.001 | 0.054 | 0.021 | 0.344 | 0.002 |
| Species × Tissue | <0.001 | <0.001 | 0.016 | <0.001 | <0.001 | <0.001 | <0.001 | 0.598 | <0.001 | 0.009 | 0.001 |

AUR, acid-unhydrolysable residue.

### 3.2. Effects of Different Treatments on the Mass Loss of Four Substrates

For absorptive roots litter, +N significantly inhibited the early-stage mass loss of *Pm* and early- and later- stage mass loss of *Ss* (Figure 2c,d and Tables 3 and S1). +P significantly stimulated the root decomposition of the two species in the late stage (Figure 2c,d and Tables 3 and S1). The +NP treatment had a negative effect on the *Ss* root decomposition in the late stage (Figure 2c,d and Tables 3 and S1). For the leaf litter, +N significantly stimulated the early-stage decomposition of *Pm* (Figure 2a,b and Table S1). Compared to the CK treatments, the foliar mass losses in both species were significantly higher in the +P and +NP treatments throughout the decomposition process ($p < 0.05$), except for the early-stage decomposition of *Ss* leaf litter in the +NP treatment (Figure 2a,b and Tables 3 and S1).

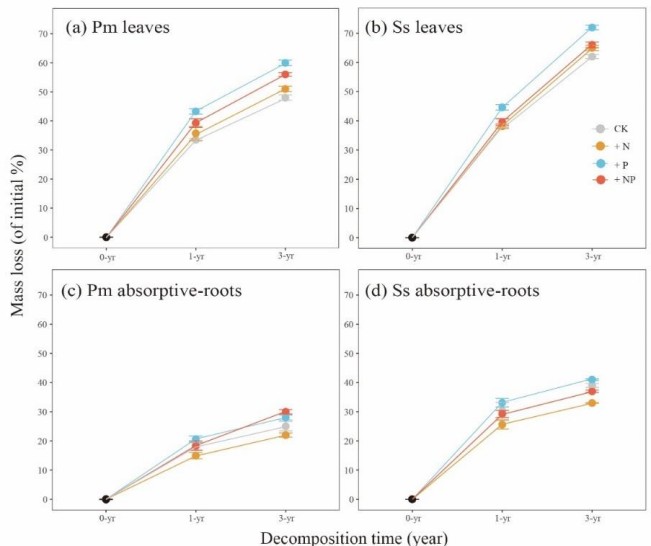

**Figure 2.** Mass loss of four substrates under different treatments in the 1-year and 3-year stages of decomposition. Significant differences between the means were determined using Tukey's honestly significant difference test ($p < 0.05$). CK, control; +N, nitrogen addition; +P, phosphorus addition; +NP, combined additions of nitrogen and phosphorus; *Pm*, *Pinus massoniana*; *Ss*, *Schima superba*.

**Table 3.** Results (*p*-values) showing the effects of nutrient addition on the 3-year decomposition of leaves and absorptive roots of *Pinus massoniana* and *Schima superba*.

| Substrates | +N vs. CK | | | +P vs. CK | | | +NP vs. CK | | |
|---|---|---|---|---|---|---|---|---|---|
| | **Effect** | ***F*** | ***P*** | **Effect** | ***F*** | ***P*** | **Effect** | ***F*** | ***P*** |
| *Pinus massoniana* leaves | *ns* | 5.30 | 0.061 | + | 83.99 | <0.001 | + | 53.98 | 0.000 |
| *Pinus massoniana* absorptive roots | *ns* | 5.55 | 0.057 | + | 14.56 | 0.009 | *ns* | 4.89 | 0.069 |
| *Schima superba* leaves | *ns* | 3.24 | 0.122 | + | 95.35 | <0.001 | + | 11.86 | 0.014 |
| *Schima superba* absorptive roots | − | 88.35 | <0.001 | + | 8.38 | 0.028 | − | 7.01 | 0.038 |

CK, control; +N, nitrogen addition; +P, phosphorus addition; and +NP, combined additions of nitrogen and phosphorus. For effect, '+', '−', and '*ns*' indicate positive, negative, and no effect of the nutrient addition on the decomposition compared with CK, respectively.

### 3.3. Carbon Fractions and Microbial Enzymatic Activity

Compared with the CK treatment, +N significantly increased the residual AUR concentration in *Pm* and *Ss* roots and increased the residual cellulose concentration in *Ss* roots throughout the decomposition process (Figure 3 and Table S2). +P had a negative effect on the residual cellulose and hemicellulose concentrations in *Pm* roots, and +NP had a positive effect on the residual AUR concentration in *Ss* roots during late-stage decomposition (Figure 3). For leaf litter, the residual AUR, cellulose, and hemicellulose concentrations had no significant differences among the four treatments throughout the decomposition process (Figure 3 and Table S2).

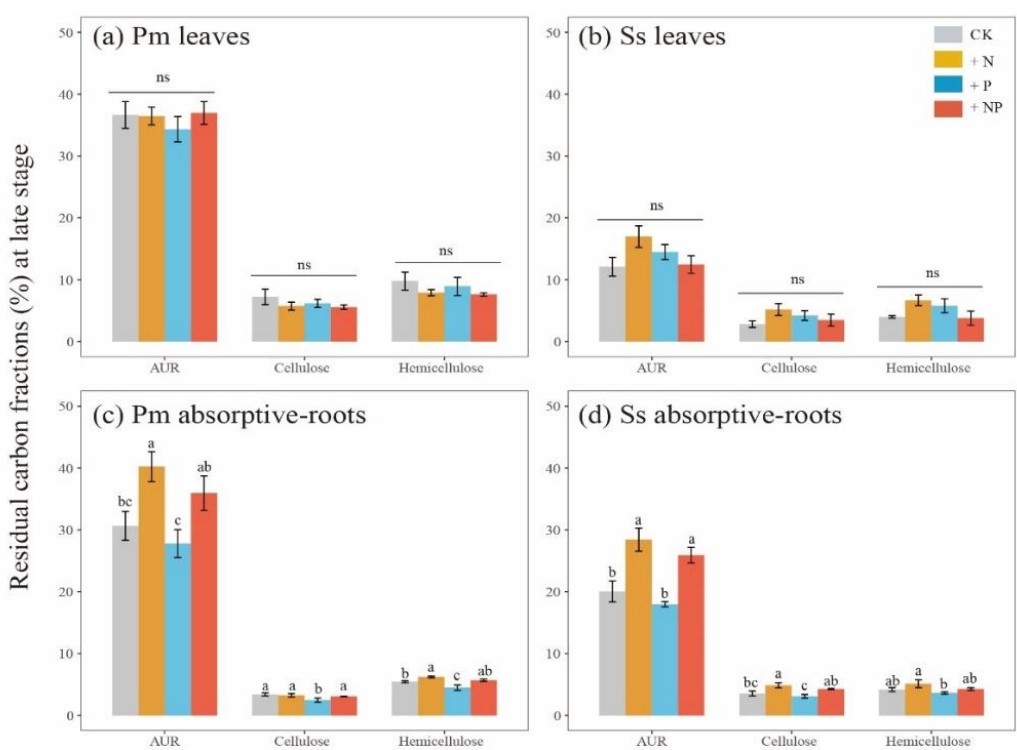

**Figure 3.** Concentrations of residual carbon fractions in four substrates under different treatments in the 3-year stage of decomposition. Carbon fractions include AUR, cellulose, and hemicellulose. Values are the mean ± SE (*n* = 4). Significant differences between means were determined using Tukey's honestly significant difference test. Different letters indicate significant differences among treatments (*p* < 0.05). CK, control; +N, nitrogen addition; +P, phosphorus addition; +NP, combined additions of nitrogen and phosphorus; AUR, acid-unhydrolysable residue; *Pm, Pinus massoniana; Ss, Schima superba*.

For the *Pm* leaf litter, the levels of the four hydrolytic enzymes were significantly higher during the early stage of decomposition under the +N treatment than under the CK treatment (Table S3). For the *Ss* leaf litter, +P treatment significantly decreased the AP activity in the early stage of decomposition (Table S3), and the +P and +NP treatments significantly increased the NAG, CBH, and AP activities in the late stage of decomposition compared with the CK treatment (Figure 4). For the root litter, the +NP treatment in *Pm* increased throughout the decomposition process, while the +N and +NP treatments decreased the AP activity in *Ss* during the late stage of decomposition (Figure 4 and Table S3). The +N and +NP treatments significantly decreased the oxidase activity in *Pm* and *Ss* roots during the late stage of decomposition. The nutrient addition had no effect on PPO and PER activities in the *Ss* leaf litter but had a negative effect on PER activity in the *Pm* leaf litter (Figure 5).

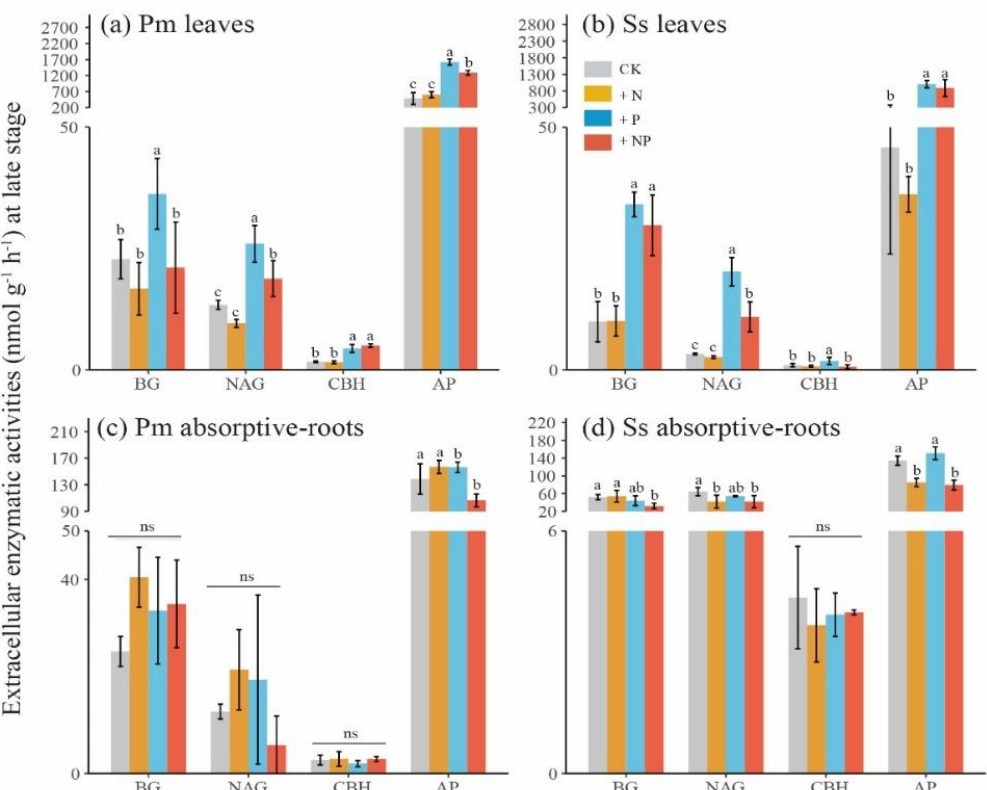

**Figure 4.** Activity of four microbial hydrolase extracellular enzymes in four substrates under different treatments in the 3-year stage of decomposition. Values are the mean ± SE (*n* = 4). Significant differences between means were determined using Tukey's honestly significant difference test. Different letters indicate significant differences among treatments (*p* < 0.05). CK, control; +N, nitrogen addition; +P, phosphorus addition; +NP, combined additions of nitrogen and phosphorus; BG, β-1,4-glucosidase; NAG, β-1,4-N-acetylglucosaminidase; CBH, cellobiohydrolase; AP, acid phosphatase; Pm, *Pinus massoniana*; Ss, *Schima superba*.

### 3.4. Relationships among Mass Loss, AUR Concentration, and Enzymatic Activity

The absorptive root mass loss was significantly correlated with the AUR concentration, whereas leaf decomposition was not (Figure 6). Specifically, the root mass loss was significantly related to microbial oxidase activity during the late stages (Figures 5 and 6). Foliar mass loss was significantly related to microbial hydrolytic enzyme activities in both the early and late stages of decomposition (Figure 7).

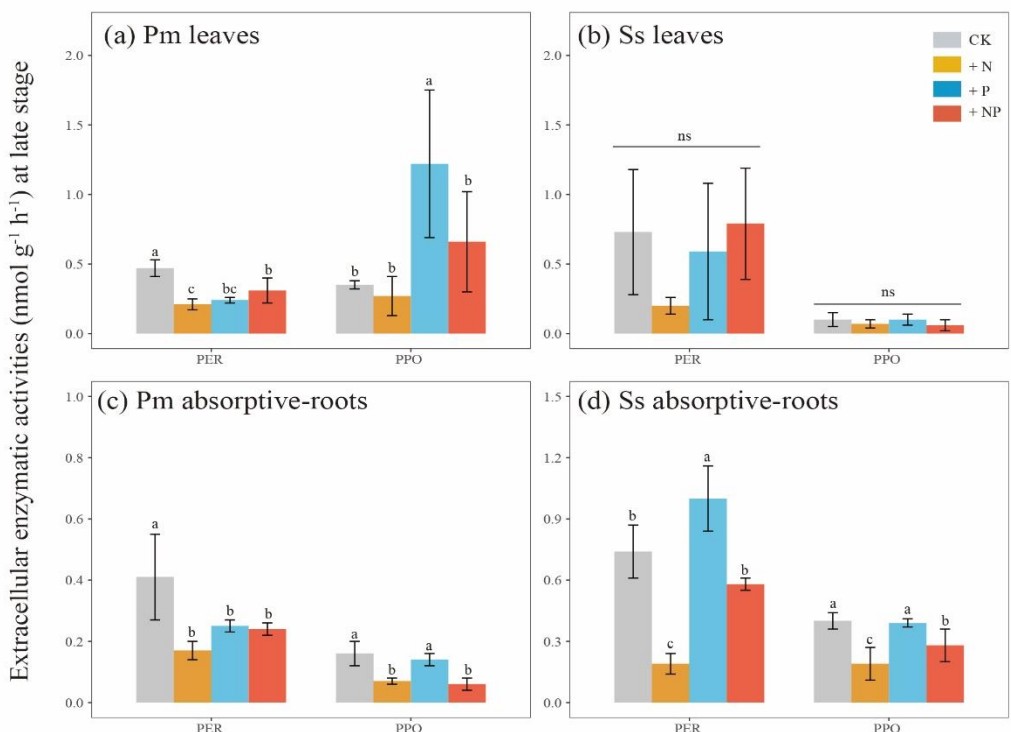

**Figure 5.** Activity of two microbial oxidase extracellular enzymes in four substrates under different treatments in the 3-year stage of decomposition. Values are the means ± SE (*n* = 4). Significant differences between means were determined using Tukey's honestly significant difference test. Different letters indicate significant differences among treatments (*p* < 0.05). CK, control; +N, nitrogen addition; +P, phosphorus addition; +NP, combined additions of nitrogen and phosphorus; PER, peroxidase; PPO, polyphenol oxidase; Pm, *Pinus massoniana*; Ss, *Schima superba*.

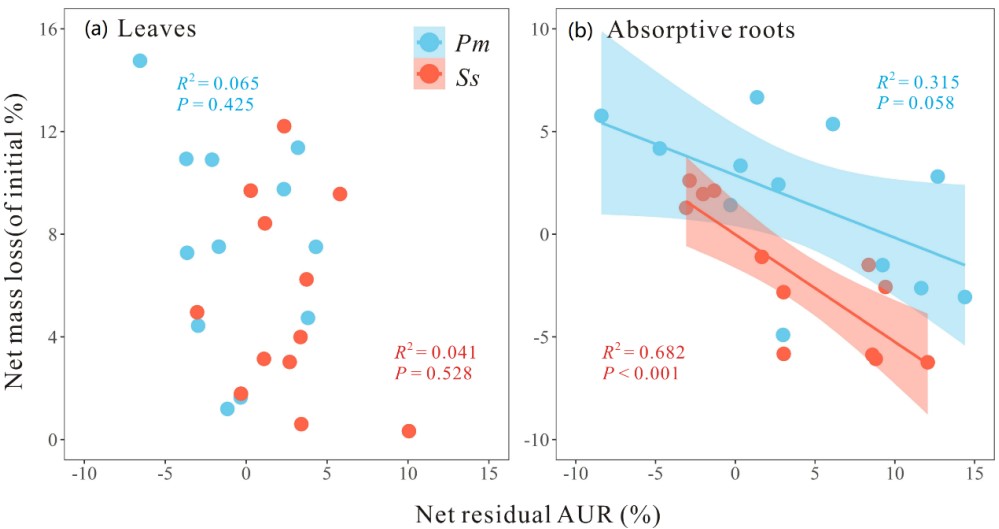

**Figure 6.** Relationships between the net residual AUR concentration and net mass loss for decomposing leaves and absorptive roots in the 3-year stage of decomposition. 'Net values' were calculated by deducting the residual AUR concentration or mass loss of the control treatment from that of the nutrient-addition treatments. Different nutrient treatments were pooled, *n* = 12, *p* < 0.05. AUR, acid-unhydrolysable residue.

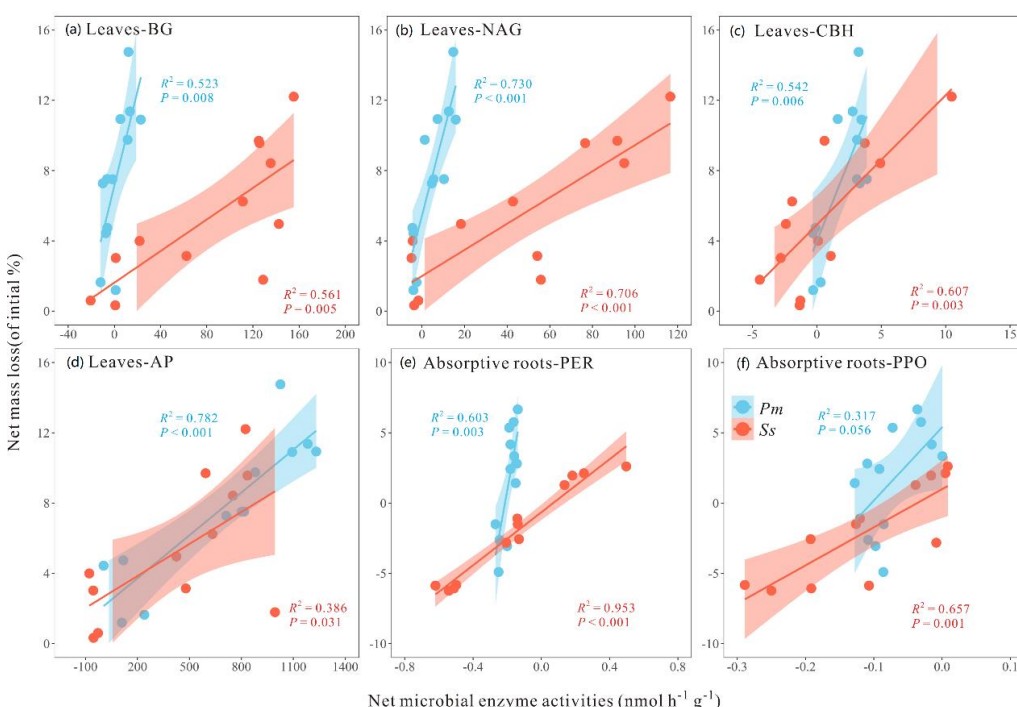

**Figure 7.** Relationships between the net microbial enzyme activity and net mass loss for decomposing leaves and absorptive roots in the 3-year stage of decomposition. 'Net values' were calculated by deducting the enzyme activity or mass loss of the control treatment from that of the nutrient-addition treatments. Different nutrient treatments were pooled, $n = 12$, $p < 0.05$. BG, β-1,4-glucosidase; NAG, β-1,4-N-acetylglucosaminidase; CBH, cellobiohydrolase; AP, acid phosphatase; PER, peroxidase; PPO, polyphenol oxidase.

## 4. Discussion

Previously, we reported the negative effects of simulated N deposition on absorptive root mass loss in the early stages of decomposition (after 1 year) at this experimental site [9]. Using the same set of samples, we sought to understand whether this effect persisted after 3 years. Consistent with our first hypothesis, the decomposition rates of the absorptive roots after 3 years were significantly or marginally lower under the +N treatment than under the CK treatment. The inhibited decomposition of roots after 3 years in the presence of N could be related to the quality of the substrate, such as a high concentration of AUR/lignin or polyphenols [8,31], which can easily combine with soil inorganic N ions to form recalcitrant compounds [13,32]. The residual AUR concentration in the absorptive roots was significantly higher after 3 years of decomposition under the +N treatment than under the CK treatment. In addition, we found positive relationships between the net residual AUR and net mass loss of root litter after this period. These findings indicate that the chemical mechanisms still play a dominant role in the underlying inhibitory effect of N deposition on root decomposition in the late stage of decomposition, as was observed for the early stage.

Except the chemical mechanism, a microbial mechanism may play an important role in underpinning the negative effect of an exogenous N addition on root decomposition. In our study, oxidase (PER and PPO) activities were significantly reduced in the +N treatment compared to that in the CK treatment. These results were consistent with the previous research showing that the suppressive effects of a N addition on diameter-based/branch-order fine root decomposition are closely related to decreases in the peroxidase and phenol oxidase activities [33,34]. The degradation of lignin/AUR are mainly driven by white rot fungi, which can secrete oxidases and peroxidases to catalyse the cleavage of carbon-carbon bonds and ether linkages [35,36]. Additionally, the mass loss of roots after 3 years was also significantly related to oxidase activities, demonstrating that the suppression of lignin-

degrading enzymes also contributes to the inhibitory effect of a N addition on absorptive root decomposition. Thus, our findings suggest that chemical and microbial mechanisms jointly contribute to the suppressive effect of a N addition on absorptive roots during late-stage decomposition.

Partly inconsistent with our first hypothesis, we found that the N addition did not change the decomposition rate of the leaves at the later stage. This may be related to interactions between the abiotic and biotic, such as N enrichment, litter quality and the enzyme activities of microbial degradation [37,38]. It is widely accepted that low-quality litter has lower decomposition rates under higher N deposition conditions (larger N enrichment) based on the microbial N-mining theory [1,39]. However, we found that the N addition did not decrease the *Pm* and *Ss* leaf decomposition, even though *Pm* leaf litter had a lower initial AUR content than *Ss* leaf litter. This highlights that the initial lignin/AUR content might not always explain the effects of a N addition on litter decomposition [37,38]. Additionally, the lignin/AUR enrichment during late stage may limit the shift from lignin-encrusted carbohydrates to hydrolases due to the microbial N-mining effect [39,40]. This may weaken the energy source available for microbial productivity [40]. Interestingly, the PER activity in the soil under the *Pm* leaf litterbags was significantly decreased in the N addition compared to the CK treatment. However, there was not a significant effect of N addition on leaf litter decomposition, which may be related to the contrasting effects of nutrient deposition on plant litter degradation and soil organic matter storage [41].

In contrast to the effects of N addition on leaf litter decomposition, the P addition consistently stimulated leaf litter decomposition. This positive effect is likely related to the growth of roots and soil P limitations in subtropical forests [5,42,43]. Generally, microbes acquiring P via the decomposition of organic matter are suppressed when there is a sufficient P supply [44]; however, an exogenous P addition may not always meet the microbial needs, particularly as the decomposition progresses. In the later stages of decomposition, the leaf substrates in the litterbags primarily contained recalcitrant chemical components [3]. In addition, fresh live roots around the leaf litterbags could easily mobilise inorganic P ions via enhancing the activity of soil microorganisms [45], especially under the +P treatment, for example, and contribute to the release of large amounts of acid phosphatases into the litter layer [46]. This results in a persistent P limitation of microbial decomposers in the litter [47,48]. We found that AP activities in the leaf litter of two species were significantly increased in the late stage of decomposition, which further indicated that the P limitation was the main factor driving leaf litter decomposition, i.e., the microbial mechanism still underlying leaf litter decomposition during the later stage.

Consistent with the positive effects of a P addition on leaf litter decomposition in the late stage, +P stimulated the later absorptive root decomposition. Interestingly, the P addition did not influence the early root decomposition. The hysteresis effect of the P deposition on root decomposition may be attributed to two factors. First, the effects of an exogenous P addition on early litter decomposition may be limited by the soil-buffering effects [49] or immobilization by microorganisms in the leaves [50,51] in P-limited forests. Second, although there is continuous P input into the soil, this exogenous P may be first used by live roots or microbes, especially mycorrhizal fungi [52], for growth, leading to a persistent P-limitation effects on microorganisms for root litter decomposition. Coincidentally, the AP activities of the absorptive roots of *Pm* and *Ss* were significantly or marginally higher under the +P treatment than under the CK treatment, to some extent indicating that P was still the limited element during the later stage decomposition of the roots.

We expected that both the addition of N and P would have neutral effects on aboveground and belowground litter decomposition, because the addition of these nutrients individually had contrasting effects, i.e., positive vs. negative, respectively. However, the effects of the +NP treatment varied with the substrate type. In the late stage, the +NP treatment increased the leaf litter decomposition but decreased the absorptive root decomposition. The positive effects of the +NP on leaf litter decomposition were consistent with

the effects of the +P treatment, with higher AP activities under the +NP treatment than under the CK treatment, similar to the patterns under the +P treatment. These results imply that microbial mechanisms play a key role in leaf decomposition, even with a N input. By contrast, the negative effects of the +NP treatment on root decomposition were in line with the negative effects of the +N treatment. This indicates that, in the late stage, the chemical mechanisms (i.e., binding of inorganic N ions to recalcitrant compounds) dominated over the microbial mechanisms (i.e., microbial enzymatic activities) in the decomposition of roots under the +NP treatment. Notably, our study took the microbial enzymes as the index to represent the microbial mechanisms, while the microbial community structure was also an important factor in regulating the effects of nutrient deposition on litter decomposition [53,54]. Consequently, future studies should pay particular attention to the microbial mechanisms when exploring the responses of root and leaf decomposition to long-term N and P additions for expanding our understanding of C and nutrient cycling in the context of global changes [48,55].

## 5. Conclusions

A full random experiment was conducted to examine the responses of leaves and absorptive roots to nutrient deposition (control, +N, +P, and +NP) in *P. massoniana* and *S. superba* forests in subtropical China during early- (1-year) and late-stage (3-year) decomposition. We found that (1) the +N treatment decreased the early- and late-stage decomposition of absorptive roots, and the +P treatment accelerated leaf decomposition throughout the decomposition process, as well as the late-stage root decomposition. (2) The positive effects of the +NP on leaf litter decomposition were consistent with the effects of the +P treatment, while the negative effects of the +NP treatment on root decomposition were in line with the negative effects of the +N treatment. (3) The decreased decomposition rate of absorptive roots under +N was related not only an increase in the AUR concentrations but also a decrease in the oxidase enzyme activity, indicating that both microbial and chemical mechanisms contribute to late-stage root decomposition. Compared to the positive effects of the P addition on leaf litter decomposition, the P input showed hysteresis effects in accelerating the root decomposition, likely due to P limitations and biological (roots and microorganisms) immobilization in the studied subtropical forest. Overall, our findings emphasise that, despite the contrasting responses of root and leaf litter decomposition to nutrient deposition, the chemical and microbial mechanisms tend to converge over decomposition processes.

**Supplementary Materials:** The following supporting information can be downloaded at: https://www.mdpi.com/article/10.3390/f14010130/s1: Table S1 Results (*p*-values) showing the effects of nutrient addition on the early (1 year) decomposition of leaves and absorptive roots of *Pinus massoniana* and *Schima superba*. Table S2 Residual carbon fractions of leaves and absorptive roots decomposing after 3 years of decomposition since litterbag placement. Table S3 Microbial extracellular enzymatic activities of leaves and absorptive roots decomposing after 3 years of decomposition since litterbag placement.

**Author Contributions:** Conceptualisation, methodology, data curation, formal analysis, software, and writhing—original draft preparation, L.J.; writing—review and editing, visualisation, supervision, validation, and investigation, L.K.; and project administration, L.K., S.L., H.W. and L.J.; software, L.J. and J.Z.; validation, X.D., S.M., X.F., H.Y., N.M. and Y.X. All authors have read and agreed to the published version of the manuscript.

**Funding:** This research was funded by grants from the National Natural Science Foundation of China (No. 31730014, 31988102, 32071557, and 41830646) and Zhejiang A&F University Scientific Research and Development Fund Talent Initiation Project (2022LFR105).

**Institutional Review Board Statement:** Not applicable.

**Informed Consent Statement:** Not applicable.

**Data Availability Statement:** Should the manuscript be accepted, the data supporting the results will be archived in Dryad.

**Acknowledgments:** The authors acknowledge the contributions of the anonymous reviewers.

**Conflicts of Interest:** The authors have no relevant financial or non-financial interests to disclose.

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
