# Peer review of "Mechanisms Underlying Aboveground and Belowground Litter Decomposition Converge over Time under Nutrient Deposition"

_forests, doi:10.3390/f14010130_

Round 1
Reviewer 1 Report
The article is extremely interesting and relevant. I believe that the material presented, the methodological approach of the authors, will be of interest to readers.
There are a few remarks about the text of the article:
1. There is very little illustrative material in the article. Add graphs, a map of the study area in the text of the article. Do not put them in the appendix. Give a graphic diagram of the research methodology. This will allow the reader to better understand your research.
Eliminating these minor remarks will improve the quality of the article.
Author Response
Response: Thank you very much for your valuable comments. Following your suggestions, we have added a figure regarding the location of the study area and sampling sites area in the text of the article to make it clear. Please see lines 144-145 and Figure 1.
Reviewer 2 Report
It's an interesting manuscript, but it still needs improving.
Considerable editing is needed to improve readability:
However, there are several major issues with the current draft of the manuscript:
1. While this study has a limited scope because it is a regional study, such studies are still acceptable. However, authors have the option of expanding their scope by adding international literature from other countries.
2. Microbial mechanism is still unclear in current manuscript.
3. Information about the limitations of this study is missing. The authors may add clear limitations to this study.
4. Discussion: The authors should focus on major findings in the discussion. The authors should discuss their findings and avoid speculation. They should also consider removing sub-topics for the discussion to flow.
5. The current information about the conclusion is not satisfactory. Conclusions can be improved!
Round 2
Reviewer 2 Report
Overall, the authors have made substantial changes. The authors have responded to most of the comments. However, there is still a few changes that must be made in order to meet publication requirements. The coordinates used in Figure 1 cannot be read using standard view settings.
Author Response
Response: Thank you very much for your positive feedback and valuable comments. Following your suggestions, we have revised the Figure 1 to read using standard view settings. Please see lines 146-147 and Figure 1. Besides, we have throughly rechecked the grammar and references to guarantee correct in this manuscript.